# Chanakya: Learning Runtime Decisions for Adaptive Real-Time Perception

**Anurag Ghosh**[1]*  **Vaibhav Balloli**[2]*  **Akshay Nambi**[3]  **Aditya Singh**[3]*  **Tanuja Ganu**[3]

[1]Carnegie Mellon University  [2]University of Michigan  [3]Microsoft Research India
anuraggh@cs.cmu.edu, vballoli@umich.edu, aditya.singh.mat17@iitbhu.ac.in,
{akshayn, taganu}@microsoft.com

## Abstract

Real-time perception requires planned resource utilization. Computational planning in real-time perception is governed by two considerations – accuracy and latency. There exist run-time decisions (e.g. choice of input resolution) that induce tradeoffs affecting performance on a given hardware, arising from intrinsic (content, e.g. scene clutter) and extrinsic (system, e.g. resource contention) characteristics.

Earlier runtime execution frameworks employed rule-based decision algorithms and operated with a fixed algorithm latency budget to balance these concerns, which is sub-optimal and inflexible. We propose Chanakya, a learned approximate execution framework that naturally derives from the streaming perception paradigm, to automatically learn decisions induced by these tradeoffs instead. Chanakya is trained via novel rewards balancing accuracy and latency implicitly, without approximating either objectives. Chanakya simultaneously considers intrinsic and extrinsic context, and predicts decisions in a flexible manner. Chanakya, designed with low overhead in mind, outperforms state-of-the-art static and dynamic execution policies on public datasets on both server GPUs and edge devices. Code can be viewed at https://github.com/microsoft/chanakya.

## 1  Introduction

Real-time perception is an important precursor for developing intelligent embodied systems. These systems operate on sensory data on top of a hardware substrate, *viz*, a mix of resource-constrained, embedded and networked computers. Resource planning is a critical consideration in such constrained scenarios. Further, these systems need to be carefully designed to be latency sensitive and ensure safe behaviour (See Figure 2).

Many runtime execution decisions exist, e.g. the spatial resolution (scale) to operate at, temporal stride of model execution, choice of model architecture etc. The runtime decisions are influenced by (a) intrinsic context derived from sensor data (e.g. image content) (b) extrinsic context observed from system characteristics, e.g. contention from external processes (other applications). A execution framework is needed to use available context, jointly optimize accuracy and latency while taking these decisions.

Traditional execution frameworks [1, 2, 3, 4] operate in the following paradigm – optimize model to operate within a latency budget. Largely, they do not jointly optimize accuracy and latency for real-time tasks. Instead, they optimize proxies like latency budget [1], input resolution [2], latency from resource contention [4] or energy [5]. Their approach requires rule-based decision making – heuristic design for every characteristic, instead of learning the decision function. These frameworks explicitly handcraft rules to accommodate hardware capabilities. Moreover, traditional strategies (Figure 2) budgeting latency are suboptimal when operating on streaming data [6].

---

*Work done at Microsoft Research India

37th Conference on Neural Information Processing Systems (NeurIPS 2023).

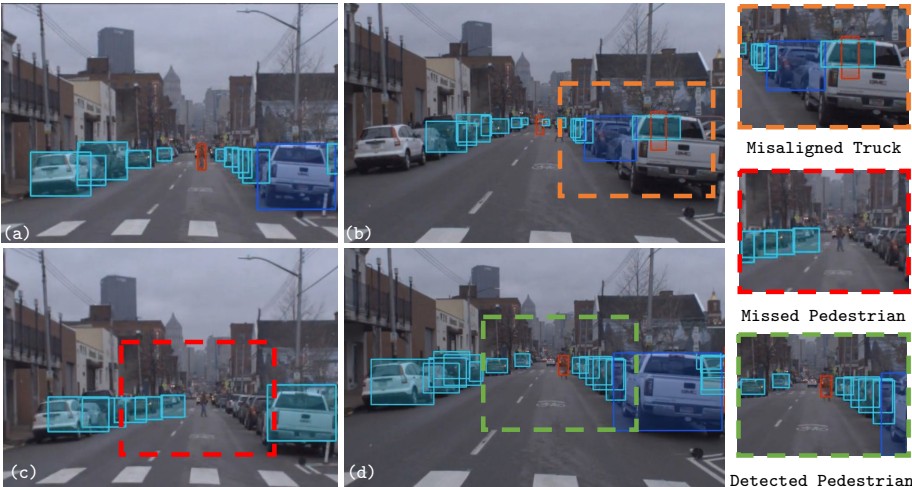

Figure 1: **Optimizing real-time perception inference.** (a) A powerful detector (HTC [7]) has very good offline (non real-time) (38.2 $mAP$) performance. (b) However, by the time it finishes execution ($750ms$ per frame), inferences are not aligned with objects (6.2 $sAP$). (c) Solely optimizing for latency via a faster model ($30ms$, RetinaNet [8] at low resolution) leads to suboptimal and unsafe behavior (6.0 $sAP$), like missing pedestrians. (d) Learning decisions with favourable inference tradeoffs with Chanakya (21.3 $sAP$) achieves better real-time performance.

Operating in the new streaming perception paradigm, we propose Chanakya[2] – a novel learning-based approximate execution framework to learn runtime decisions for real-time perception. Chanakya operates in the following manner – (a) Content and system characteristics are captured by intrinsic and extrinsic context which guide runtime decisions. (b) Decision classes are defined, such as the input resolution (scale) choices, the model choices etc. Chanakya handles these interacting decisions and their tradeoffs which combinatorially increase. (c) Accuracy and latency are jointly optimized using a novel reward function without approximating either objectives. (d) Execution policy is learnt from streaming data, and is dynamic, i.e. configuration changes during real-time execution of the perception system.

Chanakya learns performant execution policies and can incorporate new decision dimensions, stochastic system context and be ported to different hardware. Importantly, our improvements are in addition to, and complementary to models and the decision dimensions themselves – faster & more accurate models [9, 10], vision-guided runtime decisions [11, 12, 13] and system-focused decisions [14, 15] can be easily incorporated.

## 2   Related Work

**Flexible Real-Time Decision Making.** Intelligent systems that are reactive [16] and flexible [17] are well studied. Our work is related to classical resource utilization and scheduling literature [18] and theoretical frameworks for real-time reinforcement learning in asynchronous environments [19, 20]. Contrasting with prior RL based execution frameworks [3, 21] which are operating in the suboptimal traditional latency budget paradigm [6], Chanakya is situated in the streaming perception paradigm. The key challenge for all these works has been the difficulty in designing rewards to jointly optimize accuracy and latency, and previous frameworks have learnability issues (See Section 3.4). We propose Chanakya with a novel reward and asynchronous training procedure for training approximate execution frameworks on large public datasets.

**Real-time Perception and Video Analytics.** In the vision community, accuracy-latency pareto decisions or "sweet spots" for vision models have been studied [22, 6]. Drawing from these inferences, proposed design decisions exploit spatial [11, 13] and temporal [10] redundancies and improve real-

---

[2]**Chanakya** was an ancient Indian polymath, known for good governance strategies balancing economic development and social justice.

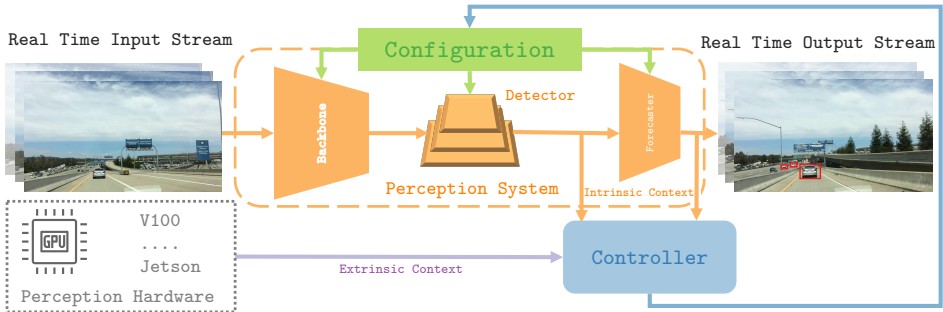

Figure 2: **High level overview.** Chanakya is a learned execution framework for real-time perception that jointly optimizes accuracy and latency. Assuming a perception system operating on a real-time video stream, we describe meta-parameters that are modified during execution – they depend on the scene (intrinsic context) and hardware considerations (extrinsic context). A specific choice of meta-parameters corresponds to a runtime execution configuration decided by the learned controller. The controller emits a configuration given context so that real-time accuracy is maximized.

time perception. These works are complementary to our work, and their meta-parameters can be learned via Chanakya, and changed at runtime (static in their work). Improved models [23, 24, 9], say via found by architecture search [25, 26] during training, compression techniques like quantization [27] and pruning [28] are also complementary.

In the systems community, Chanakya is related to video analytics efforts for the edge and cloud, wherein works aim to optimize design decisions [15, 29, 14, 30, 31] during training and inference. Chanakya can incorporate their decision designs during inference (like AdaScale [2]). For example, LegoDNN [15] decides which NN blocks to use via a novel optimization formulation, which instead can be learned via Chanakya. Mistify and Ekya [14, 31] tailor models considering offline accuracy, latency and concept drift and these model choices can be selected by Chanakya at runtime.

**Approximate Execution Frameworks.** Many approximate execution frameworks have been designed for real-time perception. However, instead of maximizing real-time performance, they optimize proxies such as energy [5], latency from contention [4], or latency budget [1] and employ handcrafted rule-based heuristics. Previous works [4] have also highlighted the need for reinforcement learning (RL) based methods to learn runtime decisions. Chanakya instead learns runtime decisions to maximize streaming performance conditioned on intrinsic and extrinsic context, operating on a variety of decision dimensions. Many systems frameworks describe runtime decisions such as changing input resolution [2, 32], switching between models [1, 32], changing model configurations with latency [4], selecting frames to execute models vs trackers [33, 5], deciding between remote execution on the cloud and local execution on the edge [33, 34, 35]. These decisions can be learnt by Chanakya.

## 3 Chanakya: Learned Execution Framework for Real-Time Perception

### 3.1 Background on Streaming Perception

Traditionally, for real-time perception, model is executed at input video's frequency, say, 30 FPS. Thus the model is constrained to process a frame within a latency budget, in this case, 33 ms per frame. Streaming perception [6] instead places a constraint on the algorithm's output, no constraints are placed on the underlying algorithm. The constraint is – *anytime the algorithm is queried, it must report the current world state estimate at that time instant.*

**Formal Definition:** Input stream comprises of sensor observation $x$, ground truth world state $y$ (if available) and timestamp $t$, denoted by $\{O_i : (x_i, y_i, t_i)\}_{i=1}^T$. Till time $t$, algorithm $f$ is provided $\{(x_i, t_i)|t_i < t\}$. Algorithm $f$ generates predictions $\{(f(x_j), s_j)\}_{j=1}^N$, where $s_j$ is the timestamp when a particular prediction $f(x_j)$ is produced. This output stream is *not* synchronised with input stream and is not necessarily operating on every input. Real-time constraint is enforced by comparing the most recent output $f(x_{\varphi(t)})$ to ground truth $y_i$, where $\varphi(t) = \operatorname{argmax}_j s_j \leq t$.

**Throughput vs Latency.** Assuming two hardware setups, one with a single GPU available and another with infinite GPU's available (simulated), we employ a Faster R-CNN detector (no forecaster)

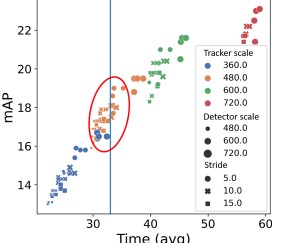

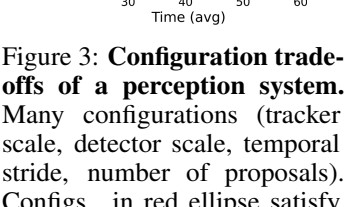

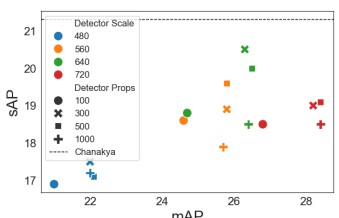

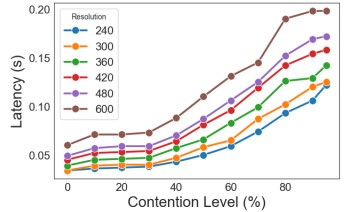

Figure 3: **Configuration trade-offs of a perception system.** Many configurations (tracker scale, detector scale, temporal stride, number of proposals). Configs. in red ellipse satisfy real-time budget $33ms$.

Figure 4: **Offline vs Real-time Perception.** mAP vs sAP for different configurations of same system. An increase in offline performance (mAP) doesn't imply an increase in real-time performance (sAP).

Figure 5: **Extrinsic Factors.** Process contention is dynamic, and induces latency differences for a detector (Faster-RCNN) at a given scale. Notice that this relationship is non-linear. Other components follow similar trends.

on a P40 GPU. The sAP scores obtained are 13.9 sAP and 15.7 sAP respectively. Additional compute (GPU's) improves system throughput, but does not improve latency of real-time execution. Thus it is not a solution for improving real-time perception.

## 3.2 High Level Overview

We design a learned execution framework for real-time perception that jointly optimizes accuracy and latency. We consider object detection to be the instantiation of our perception problem, however, the described principles apply without loss of generality.

Consider real-time object detection. The real-timeness requirement of streaming perception is fulfilled by pairing our detector (say Faster R-CNN) with object tracker (Tracktor [36]) and state estimator (Kalman Filter). We can change meta-parameters like detector's and tracker's scale, temporal stride of tracker along with internal parameters (for instance, number of proposals in the detector) and generalize this to include the choice of components themselves (the detector/tracker etc). Consider these decisions to be configuration of the algorithm $f$, and a policy $\pi$ that emits this configuration. We wish to learn a controller that governs the execution policy $\pi^\star$, which is triggered periodically to decide the runtime configuration. In this work, we assume that all the other components (detector, tracker etc) have been trained already and we solely learn the controller.

We formulate learning the runtime decision as a deep contextual bandit problem, where a sequence of independent trials are conducted and in every trial the controller chooses a configuration (i.e. action) $a$ from the set of possible decisions $\mathbb{D}$ along with the provided context $z$. After a trial the controller receives a reward $r$ as feedback for the chosen configuration and updates the execution policy. As we will see in Section 3.3, intrinsic and extrinsic factors affect policy $\pi$. Thus we define context $z$ encapsulating these factors which is provided to the controller. We learn a $Q$-function $Q_\theta(z, a)$, for every action $a$ given context $z$, where $Q$-values define the policy $\pi$ to configure $f$.

## 3.3 Intrinsic and Extrinsic Context Guide Runtime Decisions

Numerous factors inform runtime decisions in real-time perception. Figure 3 depicts configurations for a predefined perception system. Randomly picking a "good" configuration (marked by red ellipse) can result in a sub-optimal decision, causing a drop of at least 3 mAP in comparison to the "optimal" configuration. Optimal decision shifts with hardware change, on Argoverse-HD using a P40 GPU, the *best* scale turns out to be 600 instead of 900 on a 1080Ti GPU [6]. Selection of algorithm components (say, the detector) is itself a decision with trade-offs [22]. It should be noted that while recent models [10] satisfy the streaming perception constraint, they are optimized for specific hardware [13]. Detection models exhibit different kinds of errors [37, 38], and changing the scale affects their performance. Lastly, offline and real-time perception performance are not correlated (Figure 4).

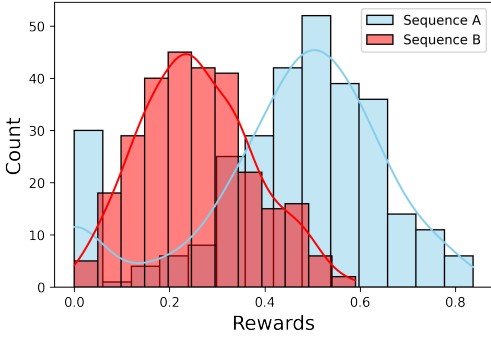 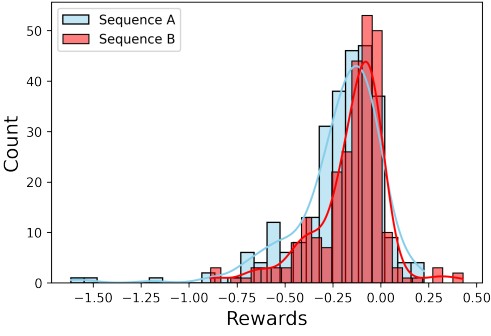

(a) $R$ for sequences $A$ and $B$.      (b) $R_{fixed\_adv}$ for sequences $A$ and $B$.

Figure 6: **Understanding** $R_{fixed\_adv}$. In many video sequences, performance measure (single frame loss $L$) is high for all configurations (e.g. all objects are large, highly visible), such as Sequence A. Compared to sequences (like B) where performance is lower and more heavily dependent on detector type and scale (e.g. objects are small or occluded). $R_{fixed\_adv}$ corrects this bias.

Intrinsic factors like image content influence decisions – static configurations are sub-optimal. E.g., We can optimize scale at runtime [2] based on video content to improve throughput while maintaining accuracy. Extrinsic factors introduce additional challenges. Optimal decisions change with the stochastic nature of the external process, such as resource contention induced by other running processes. Figure 5 shows change in latency with contention levels and this relationship is non-linear. Other such extrinsic factors include network conditions and power considerations. Extrinsic context are non-differentiable, thus we do not use an end-to-end network to learn the overall context $z$.

We concatenate outputs of the following functions to form the overall context to the controller,

**1. Switchability Classifier:** Models exhibit different error characteristics, motivating the need to switch models at runtime [1]. Switchability classifier guides model switching at runtime.

**2. Adaptive Scale Regressor:** AdaScale [2] predicts the optimum scale an image should be inferred at, such that *single frame latency* is minimized. We use their regressor as an intrinsic context.

**3. Contention Sensor:** Process Contention affects model latency. We employ the GPU contention generator and sensor from [4], simulating discrete contention levels.

**4. Scene aggregates:** We also compute frame-level aggregates: (i) **Confidence Aggregates**: We compute the mean and the standard deviation of the confidence value associated with the predictions. The intuition is if the average confidence is low, which might indicate execution of an "expensive" configuration. (ii) **Category Aggregates**: We count of instances of each category in the predictions. This might capture *rare* objects, which may require additional resources to be localized. (iii) **Object Size Aggregates**: We categorize and aggregate predictions as small, medium and large detections (like COCO thresholds [39]). This is useful as small objects are considered difficult to detect.

### 3.4 Reward Jointly Optimizes Latency And Accuracy

Traditionally, output $\hat{y}_i$ is emitted for observation $O_i$ at time $\hat{t}_i$ ($> t_i$), and the reward [3, 21] is defined as

$$R_{trad} = \phi(y_i, \hat{y}_i) + \lambda\xi(t_i, \hat{t}_i)$$

considering accuracy via $\phi$ and latency via $\xi$ separately. However, hyperparameter $\lambda$ for this additive reward is globally inconsistent (i.e. changes depending on intrinsic/extrinsic context) which doesn't work for large datasets. In our experiments we observed that controller collapsed to predict a single configuration when rewards of this form were provided (depending on $\lambda$), and did not learn.

Now, consider the streaming perception setup described in Section 3.1. Deriving from this paradigm, our rewards optimize accuracy and latency simultaneously. Intuitively, reward $R$ compares the model output with a ground truth stream of the actual world state. Assume algorithm $f$ is governed by policy $\pi$ receives observation $O_x$, controller performs action $a_x$ at time $t_x$ to configure $f$ and another

**Algorithm 1:** Obtaining Observations From Streaming Perception System

Initialize policy $\pi$
Set change system configuration probability $p$
$\tau = 0$
**for** $e = 0, 1, \cdots, E - 1$ **do**
  Reset simulator;
  **for** *every sequence* **do**
    Initialize empty stream replay buffer S;
    $i = 1$;
    **while** *streaming* **do**
      // System executes on streaming data
      ...
      $value = $ Draw from $Bernoulli(p)$ ;
      **if** *value == true* **then**
        Observe $x_i$ from the stream;
        $z_\tau = context(x_i)$;
        $a_\tau = \pi(z_\tau)$ ;
        Change system config. using $a_\tau$;
        $t_{a_\tau} = time.now()$;
        Store $(z_\tau, a_\tau, t_{a_\tau})$ in stream replay buffer S ;
        $\tau = \tau + 1$;
      $i = i + 1$;
    $\pi = train\_controller(\pi, S)$ ;

---

**Algorithm 2:** Training the Controller

**Function** `train_controller`($\pi$: *Policy, S: Stream Replay Buffer*):
  Unpack $K$ length sequence $\{(z^1, a^1, t^1) \cdots (z^K, a^K, t^K)\}$ from stream replay buffer $S$;
  **for** $n = 1, \cdots, K$ **do**
    $r^n = R(t^n, t^{n+1})$ ;
    Store $(z^n, a^n, r^n)$ in controller's replay buffer $B$;
  $\pi = update(\pi)$ ;

**Update Policy**

---

**Function** `update`($\pi$: *Policy*):
  // Optimize the $Q$ function

  Sample $N$ tuples $\{(z^1, a^1, r^1), \cdots, (z^N, a^N, r^N\}$ uniformly from buffer $B$;
  **for** $n = 1, \cdots, N$ **do**
    // Set the targets
    $y^n = r^n$
  // Calculate the loss for the batch
  $\mathcal{L} = \frac{1}{N} \sum_{n=1}^{N} \frac{1}{|D|} \sum_{d \in a^n} [y^n - Q_\theta(z^n, d)]^2$
  Use optimizer to optimize $\theta$ to minimize $\mathcal{L}$;

---

observation $O_y$ with action $a_y$ is taken at $t_y$ ($t_y > t_x$). We propose a reward,

$$R(t_x, t_y) = L(\{y_{t_k}, \hat{y}_{\varphi(t_k)}\}_{k=x}^{y}) \tag{1}$$

where $L$ is an arbitrary single frame loss. Intuitively, the reward for action $a_x$ is streaming accuracy obtained for a stream sequence segment until a new action $a_y$ *is taken in the future* and $\varphi$ ensures only real-time results are accounted for. Reward $R(t_x, t_y)$ ensures learned controller implicitly considers real-time constraints and accuracy of the decision $a_x$, not assuming any properties of the algorithm $f$ or loss $L$. Thus, it can applied to other perception tasks.

While [6] defined $L_{streaming} = L(\{y_{t_k}, \hat{y}_{\varphi(t_k)}\}_{k=0}^{N})$ and corresponds to $R(0, T)$ (Loss over N observations from a video stream of length T), $L_{streaming}$ was employed to characterize performance of $f$ on the sensor stream. Directly employing it as reward made training the controller sample-inefficient, and did not converge before we exhausted our training budget (of 10 epochs).

While reward proposed in Equation 1 is learnable, it is biased towards the inherent characteristics of the video sequence. Consider two video sequences $A$ and $B$, Sequence $A$ with one large object that can be easily detected would have much higher average rewards than Sequence $B$ with many occluding and small objects. To normalize for this factor, we consider a fixed policy $\pi_{fixed}$, i.e., a fixed configuration of algorithm $f$ and propose,

$$R_{fixed\_adv}(t_x, t_y) = R_\pi(t_x, t_y) - R_{\pi_{fixed}}(t_x, t_y) \tag{2}$$

where $R_{\pi_{fixed}}$ is the reward obtained by executing algorithm $f$ following a fixed policy. To implement $R_{fixed\_adv}$, $f$ with fixed policy $\pi_{fixed}$ are executed and prefetched the predictions and the timestamps are stored. During training, we compute $R_\pi$ and load $\pi_{fixed}$ outputs to simulate $R_{\pi_{fixed}}$.

This eliminates the bias in the inital proposed reward, and provides the *advantage* of learned policy over fixed policy (See Figure 6). In Section 4.1, we show any fixed policy configuration ($\pi_{fixed}$) suffices and the policy learned is resilient to variations of this selection. We refer to the reward in Equation 1 as $R_1$ and Equation 2 as $R_2$ in our experiments for simplicity.

### 3.5 Controller Design Decisions

Our controller design decisions (such as employing deep contextual bandits) are motivated by severe real-time constraints. For e.g., we considered individual frames as independent trials while learning our controller, i.e. we compute context from a single frame to incur minimal overhead (amortized overhead of around 1%). Considering sequences as controller's input incurred significant overheads and impeded performance.

Table 1: **Improvements on a predefined system.** Chanakya outperforms competing execution policies for a predefined system. All the execution policies operate on top of a perception system employing the same components: Faster R-CNN and Kalman Filter.

| Approach | $sAP$ | $sAP_{50}$ | $sAP_{75}$ |
|---|---|---|---|
| 1. Streamer (s=900) [6] (Static Policy) | 18.2 | 35.3 | 16.8 |
| 2. Streamer (s=600) [6] (Static Policy) | 20.4 | 35.6 | 20.8 |
| 3. Streamer (s=600, np=300) (Static-Expert Policy) | 20.8 | 36.0 | 20.9 |
| 4. Streamer (s=600, np=500) (Static-Expert Policy) | 20.9 | 35.9 | 20.9 |
| 5. Streamer + AdaScale [2] (Dynamic-Traditional Policy) | 13.4 | 23.1 | 13.8 |
| 6. Streamer + AdaScale + Our Scheduler (Dynamic-Trad. Policy) | 13.8 | 23.4 | 14.3 |
| 7. **Chanakya** ($s_1$, $np_1$, **R**=$R_1$) | 21.0 | 36.8 | **21.2** |
| 8. **Chanakya** ($s_1$, $np_1$, **R**=$R_2$, $\pi_{fixed}$=(s=480, np=300)) | **21.3** | **37.3** | 21.1 |
| 9. Offline Upper Bound (s=600, Latency Ignored) | 24.3 | 38.9 | 26.1 |

**Combinatorially Increasing Configurations.** Consider decision space $\mathbb{D} = \{D_1, D_2..D_m\}$, with dimensionality $M = |\mathbb{D}|$, where each decision dimension $D_i$ corresponds to a configuration parameter of the algorithm $f$ discretized as $d_k$. Thus, an action $a$ has $M$ decision dimensions and each discrete *sub-action* has a fixed number of choices. For example, if we consider 2 decision dimensions, $\mathbb{D} = \{D_{scale}, D_{model}\}$, the potential configurations would be $D_{scale} = \{720, 600, 480, 360, 240\}$, $D_{model} = \{yolo, fcos, frcnn\}$. Using conventional discrete-action algorithms, $\prod_{d\in\mathbb{D}} |d|$ possible actions need to be considered. Efficiently exploring such large action spaces is difficult, rendering naive discrete-action algorithms like [3, 21] intractable.

Chanakya uses a multi-layer perceptron for predicting $Q_\theta(z, a_i)$ for each sub-action $a_i$ and given context $z$, and employs action branching architecture [40] to separately predict each sub-action, while operating on a shared intermediate representation. This significantly reduces the number of action-values to be predicted from $\prod_{d\in\mathbb{D}} |d|$ to $\sum_{d\in\mathbb{D}} |d|$.

**Real-time Scheduler.** Choosing which frames to process is critical to achieve good streaming performance. Temporal aliasing, i.e., the mismatch between the output and the input streams, reduces performance. [6] prove optimality of shrinking tail scheduler. However, this scheduler relies on the assumption that the runtime $\rho$ of the algorithm $f$ is constant. This is reasonable as runtime distributions are unimodal, without considering resource contention.

Chanakya changes configurations at runtime, and corresponding $\rho$ changes. Fortunately, our space of configurations is *discrete*, and $\rho$'s (for *every* configuration) can be assumed to be constant. Thus, a fragment of the sequence where a particular configuration is selected is a scenario where shrinking tail scheduler using that runtime will hold. We cache configuration runtimes of algorithm $f$ and *modify the shrinking tail scheduler* to incorporate this assumption.

**Asynchronous Training.** Chanakya is trained in such a way that real-time stream processing is simulated, akin to challenges presented in [19, 20]. Thus, a key practical challenge is to factor in average latency for generating the context from the observations, computing the controller's action during training. We also need to ignore the time taken to train the controller as these would disrupt and change the system performance. Training the controller concurrently on a different GPU incurred high communication latency overhead.

Algorithms 1 and 2 describe the training strategy. We train the controller intermittently – after a sequence has finished processing, the controller's parameters are fixed while the sequence is streamed. While the sequence is processing, we store the $(z, a, t)$ tuples in a buffer $S$ along with all the algorithm's predictions (to construct the reward $r$). During training, first we add these tuples to controller's buffer $B$ and, sample from $B$ to update $Q_\theta(z, a)$.

**Training for edge devices.** We emulate training the controller for edge devices on a server by using the measured runtimes for each configuration on the edge device, since Chanakya only requires the runtimes of the algorithm $f_\pi$ on the target device to train the controller.

Table 2: **Handling large decision spaces.** We incorporate a tracker in our predefined perception system. Choice of detector scale and proposals, tracker scale and stride, combinatorially expands our decision space. Chanakya is able to learn a competitive policy compared to best static policies [6].

| Approach | sAP |
| --- | --- |
| Static Policy (s=900, ts=600, k=5) | 17.8 |
| Static Policy (s=600, ts=600, k=5) | 19.0 |
| Chanakya ($s_1$, $np_1$, $ts$, $k$, $R_2$) | **19.4** |

Table 3: **Learning decisions from scratch.** Chanakya is able to decide model and other meta-parameter choices even when optimal algorithmic components are unknown.

| Approach | sAP |
| --- | --- |
| Static Policy (m=$fcos$, s=600) | 16.7 |
| Static Policy (m=$yolov3$, s=600) | 20.2 |
| Static Policy (m=$frcnn$, s=600) | 20.4 |
| Chanakya (m=$m$, $s_1$, $np_1$, R=$R_2$) | **20.7** |

## 4    Experiments and Results

We analyse and evaluate Chanakya through the following experiments. We employ sAP metric [6] that coherently evaluates real-time perception, combining latency and accuracy into a single metric. It is non-trivial to compare traditional systems that use mAP and latency for evaluation as their algorithms would require major modifications. However, we do augment one such method, AdaScale [2] by incorporating Kalman Filter and our scheduler. Apart from this, we compare our approach with static policies obtained from benchmarking. Please note Chanakya's improvements are complementary and in addition to the system components themselves, newer models would lead to further improvements. All the results are reported on Argoverse-HD, unless stated otherwise.

**Improvements on a predefined system.** For a real-time perception system on a given hardware (P40 GPU), let's assume system components – detector (Faster R-CNN), scheduler (Modified Shrinking Tail) and forecaster (a Kalman Filter) are known. We optimize configurations across two decision dimensions $\mathbb{D} = \{D_s : \{360, 480, 540, 640, 720\}, D_{np} : \{100, 300, 500, 1000\}\}$, i.e., detector scale and number of proposals. We evaluate Chanakya using both variants of rewards described in Section 3.4, and only intrinsics are provided as context. We compare with best performing configurations reported by [6] (**static policy**). Further, we asked an expert who employed domain knowledge [22] to optimize the proposal decision dimension via benchmarking (**static expert policy**). We also compare with a **dynamic traditional policy**, AdaScale [2] which predicts input scale and is trained to minimize single frame latency as it is situated in traditional latency budget paradigm. Upper bound takes the best configuration of the static policy from [6] and simulate a detector with 0ms latency for every frame. Table 1 shows that we outperform all of these methods, as Chanakya can adapt dynamically (See Figure 7) via context which explains improvement over static policies and considers the real-time constraint holistically. Chanakya does not *solely* minimize single frame latency which explains improvements over the traditional policy. Similar trends hold for ImageNet-VID, please see Appendix A.

Upper bound takes the best configuration of the static policy from [6] and simulate a detector with 0ms latency for every frame.**Learning decisions from scratch.** Consider when optimal components (say, the detection model or tracker) are unknown for a given hardware. We will now introduce the third dimension, which is the choice of the model (FCOS or YOLOv3 or Faster R-CNN). Chanakya has to pick configurations across model choices, scale choices and proposal choices (if applicable, decision dimension is ignored otherwise). Using an alternative detector, i.e, FCOS and with the best scale obtained via benchmarking (paired with Streamer [6]), the static policy has an sAP of 16.7, while Chanakya has sAP of 20.7 (23% improvement). Similar performance improvement (+5.5 sAP) was also seen on ImageNet-VID.

**Handling large decision spaces.** We extend our pre-defined system (Faster R-CNN + Scheduler + Kalman Filter) with a tracker (Tracktor [36]). Tracker supports configuration of tracker scale and temporal stride, i.e. number of times it's executed versus the detector, apart from detector scale, number of proposals, increases the decision space to 500 configurations. It is not feasible to explore the entire decision space using brute-force if we consider deployment to many hardware platforms. In contrast, our approach learned the execution policy in 5 training epochs. As shown in Table 2, Chanakya learns a competitive policy that outperforms the **best static policy** from [6] by +1.6 sAP.

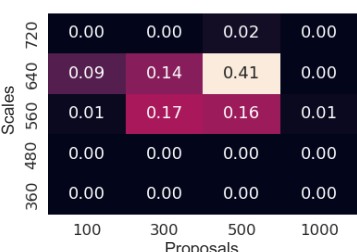

Figure 7: **Dynamic Configurations executed on P40 GPU.** For the predefined system, Chanakya changes the configuration depending on the intrinsics and achieves better performance.

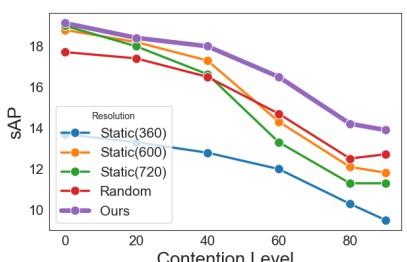

Figure 8: **Chanakya adapts to extrinsic factors.** Compared to static execution policies, Chanakya has better performance at different levels of GPU contention and degrades less at higher contention.

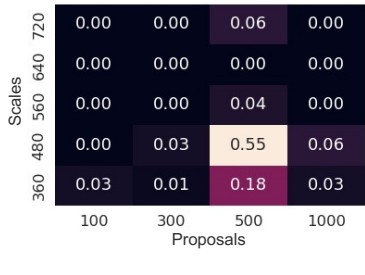

(a) Xavier NX

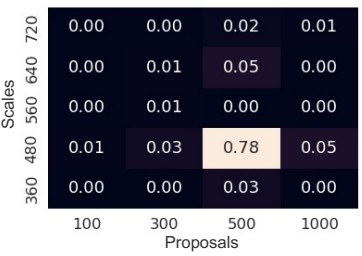

(b) Xavier AGX

Figure 9: **Configuration shift on edge devices.** The predefined system is deployed on the edge devices. Chanakya is trained via emulation, and we observe a shift in chosen configurations that matches our expectations for these hardware platforms.

**Performance and configuration shift on edge devices.** We consider deploying the predefined system discussed earlier on two edge devices – Jetson Xavier NX and Xavier AGX. We employ emulated training procedure discussed in Section 3.5. From Table 4 comparing performance on Xavier NX, Chanakya improves sAP by 12% and 28% Argoverse-HD and ImageNet-VID respectively over static policies found by benchmarking. In Figure 9, we observe shifts in configurations selected for the two devices (compared to Figure 7 evaluated on a server GPU P40). Xavier AGX is more powerful, and we observe more frequent selection of higher detection scale (s = 480) compared to Xavier NX, in line with our expectations.

**Adapting to Extrinsic Factors.** In our predefined system, we consider contention as our extrinsic factor and study Chanakya's behaviour, assuming black-box resource contention levels available at run-time. This scenario is common on smartphones/edge devices [4] where multiple processes employ the same resource. Static policies, wherein decisions are not made considering contention would degrade quickly (See Section 3.3).

Chanakya adapts to sensed contention level (provided as context) by choosing decisions which correspond to the sweet-spot for that contention level, with no additional modifications. Critically, Chanakya is flexible and adapts decisions to the contention level unlike [4] who explicitly model the problem as a rule-based system (modeling latency and accuracy with quadratic regression models) and handcraft a contention-aware scheduler. Performance degrades across all approaches with increasing contention and Figure 8 Chanakya is able to adapt better than static policies.

**New hardware and action spaces.** Consider the task of migrating a streaming perception system from a device employing P40 GPU, to newer V100 GPU. New hardware has additional features, such as faster inference at lower precision levels (P40 in comparison does offer limited support; few FP16 operations are faster). As the tradeoffs have shifted, we do not assume that components and hardware optimizations are known. We consider additional decisions – choice between inference at FP32 and FP16 and choice between models. We employed Chanakya with the following decision space:

Table 4: **Performance on edge devices.** Comparisons on Xavier-NX edge device. We can observe that Chanakya improves performance over static policies on both the datasets.

| Approach | Argoverse-HD (sAP) | ImageNet-VID (sAP) |
|---|---|---|
| Streamer (s=560, np=500) (Static Policy) [6] | 7.8 | 18.8 |
| Streamer (s=640, np=300) (Static Policy) [6] | 7.4 | 20.0 |
| Streamer with Random Selection | 7.6 | 17.5 |
| Chanakya (R = $R_1$) | 8.1 | 19.4 |
| Chanakya (R = $R_2$) | **8.3** | **22.4** |

Table 5: **Importance of Context.** All components of the considered context contribute to improved real-time performance.

| Dropped feature | sAP |
|---|---|
| None | **21.3** |
| Adaptive Scale | 19.2 |
| Category aggregates | 19.4 |
| Confidence aggregates | 20.3 |
| Object size aggregates | 20.4 |

Table 6: **Resilience.** Chanakya learns performant decisions despite using a bad fixed policy. Further, minimal performance degradation is observed between $s_1$ and $s_2$

| Approach | sAP |
|---|---|
| Ours ($s_1, np_1, R_2, \pi_{fix} = (640, 300)$) | **21.3** |
| Ours ($s_1, np_1, R_2, \pi_{fix} = (480, 300)$) | **21.3** |
| Ours ($s_1, np_1, R_2, \pi_{fix} = (360, 1000)$) | 21.1 |
| Ours ($s_2, np_1, R_1$) | 18.2 |
| Ours ($s_2, np_1, R_2$) | **20.4** |

$\mathbb{D} = \{D_m : \{resnet\_frcnn, swin\_frcnn, resnet\_cascade\_rcnn, swin\_cascade\_rcnn\}, D_p : \{FP32, FP16\}, D_s : \{800, 880, 960, 1040, 1120, 1200\}, D_{np} : \{100, 300, 500, 1000\}\}$. Without benchmarking and domain expertise, it's non-trivial to decide these meta-parameters. We consider the best known static policy (Faster R-CNN with Resnet-50 at $s = 900$) proposed by [6] for V100 GPU as our baseline which has $18.3$ $sAP$. Chanakya obtained 27.0 sAP, i.e., +8.7 sAP (47%) improvement, the highest improvement on the Streaming Perception Challenge without additional data.

### 4.1 Ablation Studies

We discuss some of the considerations of our approach using Chanakya as our perception system.

**Importance of Context.** Table 5 shows that when a context feature is dropped performance reduces by $\approx 1$ sAP. Context overheads can be viewed in Appendix C.

**Resilience.** Table 6 shows that Chanakya is resilient to the choice of the initial fixed policy $\pi_{fixed}$ used in our reward function $R_2$. Chanakya arrives at a competitive policy even if a bad fixed policy is chosen, $(360, 1000)$, which employs very small scale and the maximum number of proposals. Chanakya is resilient to the choice of action space as changing scale space from $s_1$ to $s_2$ yielded competitive performance.

## 5 Conclusion

**Limitations and Broader Impact.** While our method does not directly enable unethical applications, it might be employed to deploy and improve such applications. Further study of regulatory mechanisms for safe ML deployments and the impact of methods like Chanakya is required.

We presented Chanakya, a learned runtime execution framework to optimize runtime decisions for real-time perception systems. Our learned controller is efficient, flexible, performant and resilient in nature. Chanakya appears useful for many real-time perception applications in the cloud and edge. We plan to extend this work to scenarios wherein applications require hard performance guarantees.

## Acknowledgements

We thank Harish YVS for his insightful discussions and the reviewers for their detailed comments.

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
