Table 7: **Improvements on a predefined system. (ImageNet-VID)** All the execution policies operate on top of a perception system employing the same components: Faster R-CNN and Kalman Filter. We can observe similar trends to Argoverse-HD (Table 1).

| Approach | sAP | $sAP_{50}$ | $sAP_{75}$ |
|---|---|---|---|
| 1. Streamer ($s = 600$) [6] (Static Policy) | 34.4 | 59.8 | 35.4 |
| 2. Streamer + AdaScale [2] + Our Scheduler (Dynamic-Traditional Policy) | 32.2 | 55.5 | 33.6 |
| 3. **Chanakya** ($s = s_3, np = np_1, R = R_1$) | **37.5** | **62.0** | **40.0** |
| 4. **Chanakya** ($s = s_3, np$=$np_1, R = R_2, \pi_{fixed} = (s = 480, np = 300)$) | 37.2 | **62.0** | 39.5 |

# A    Implementation and Experimental Details

**Datasets.** We perform our experiments on the Argoverse-HD dataset [41, 6], which contains urban outdoor driving scenes from two US cities and ImageNet-VID dataset [42], which is a diverse dataset with 30 object categories.

**Performance Measure (mAP vs sAP):** Mean Average Precision or **mAP** is used to evaluate offline detection, where any detection for a frame is considered irrespective of its latency. Satisfying the formal definition above, Streaming AP or **sAP** [6] evaluates online or real-time detection, by matching the most recent detection that was provided on or before the current frame observed in a video stream. We evaluate real-time performance using the *sAP* measure, i.e., AP in the real-time setting, as it simultaneously evaluates both latency and average precision of the detections. Similar measures can be defined for other tasks like segmentation [6]. A visual walkthrough of sAP metric can be viewed in [43].

**Implementation Details.** We use the mmdetection [44] library to train and finetune our models. Further, while employing multiscale training on scales between $[720, 320]$, we follow the *1x schedule* of mmdetection and default parameters to maintain consistency. We finetune our models from a base model trained on COCO [39], for ImageNet-VID we use the protocol described in [2, 45] and for Argoverse-HD we consider every $5^{th}$ frame from the training sequences. Our controller is trained on the training set of these datasets, keeping model components frozen.

Evaluation is performed on the validation sets of ImageNet-VID and Argoverse-HD, as ground truth annotations for test set are not publically available. Unless stated otherwise, all our experiments are performed on Microsoft Azure's ND-series machines with P40 GPUs. For training, we used machines with 4 GPUs and we evaluated on a machine with one P40 GPU.

**Controller training details.** Our controller is a 4 layer MLP with the hidden layer size of 256, input size being equal to the size of our context vector (i.e., 22 for Argoverse-HD and 44 for ImageNet-VID) and output size being the sum of each action dimension, i.e. $\sum_i |D_i|$. For epsilon-greedy exploration-exploitation, we set initial $\epsilon$ as 1, decay rate as 0.999 and minimum-$\epsilon$ as 0.15 and train for 10 epochs. We train our models using both reward functions defined in Equations 1 and 2, which we refer to as $R_1$ and $R_2$, respectively.

**Tradeoffs considered.** In our experiments, we defined the following decision dimensions: (1) **Scale** ($s$)**:** $s_1 = \{720, 640, 560, 480, 360\}$ and $s_2 = \{750, 675, 600, 525, 450\}$; $s_3 = \{$ 600, 480, 420, 360, 300, 240 $\}$ (for fair comparison with AdaScale [2] on ImageNet-VID) (2) **Number of Proposals** ($np$)**:** $np = \{100, 300, 500, 1000\}$ (3) **Tracker scale** ($ts$)**:** $ts = \{720, 640, 560, 480, 360\}$ (4) **Tracker stride** ($k$)**:** $k = \{3, 5, 10, 15, 30\}$ and (5) **Model choice** ($m$)**:** $m = \{yolov3, fcos, frcnn\}$.

While adapting to newer hardware, we considered additional decisions apart from Scale ($s$) as $s_4 = \{800, 880, 960, 1040, 1120, 1200\}$ and Number of Proposals ($np$) as $np_1$. They are (1) **Models** ($m$) { $m = \{resnet\_frcnn, swin\_frcnn, resnet\_cascade\_rcnn, swin\_cascade\_rcnn\}$ (2) **Inference Precision** ($p$)**:** $p = \{FP32, FP16\}$.

**Results on ImageNet-VID** Improvements for a pre-defined system generalize to ImageNet-VID as we can observe in Table 7. From Figure 10, observe that the decisions for ImageNet-VID are different.

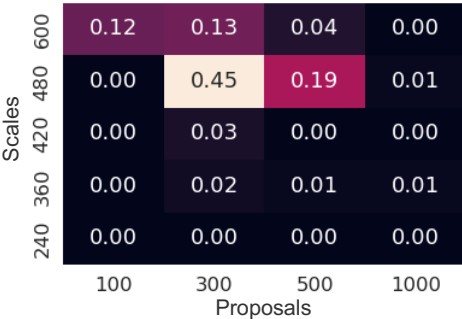

Figure 10: **Dynamic Configurations executed on P40 GPU. (ImageNet-VID)** For the predefined system, Chanakya changes the configuration depending on the intrinsics and achieves better performance. Notice that the runtime decisions are distinct from decisions taken in Argoverse-HD.

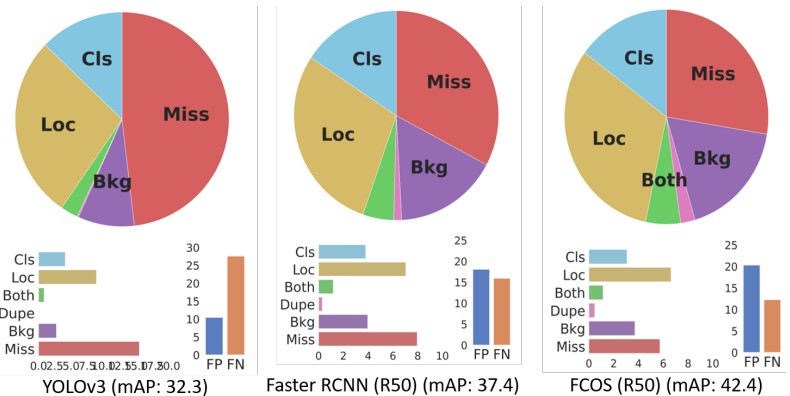

Figure 11: **Breakdown of Detection Errors on COCO using TIDE [37].** We can observe a much higher miss rate in YOLO compared to FCOS and Faster R-CNN.

## B   Additional Details on Factors Influencing Runtime Decisions

Different models exhibit different error characteristics, Figure 11 and Figure 12 show the error breakdown for COCO [39] and ImageNet-VID [42]. The error breakdowns vary across the datasets among these models, and some models are more suited for specific scenarios.

## C   Additional Details on Runtime Context

Table 8: **Switchability Classifier.** Switchability category acts as context for our controller to make a runtime decision. As we can observe, Classification performance on Argoverse-HD is far better than random selection, provided context is informative.

|  | Precision | Recall | F1-Score |
|---|---|---|---|
| Low | 0.81 | 0.87 | 0.84 |
| Medium | 0.54 | 0.56 | 0.55 |
| High | 0.71 | 0.45 | 0.55 |
| Accuracy |  |  | 0.73 |

**Switchability Classifier.** To obtain this context, during training, we execute all the supported models $F$ on image $I$ to obtain detections $\{x_{f_1}, x_{m_2}..x_{f_k}\}$, where $x_{f_i}$ is the output obtained from model $f_i$. We then compute the standard deviation of the mean IOU across all the models in $F$ on $I$ and

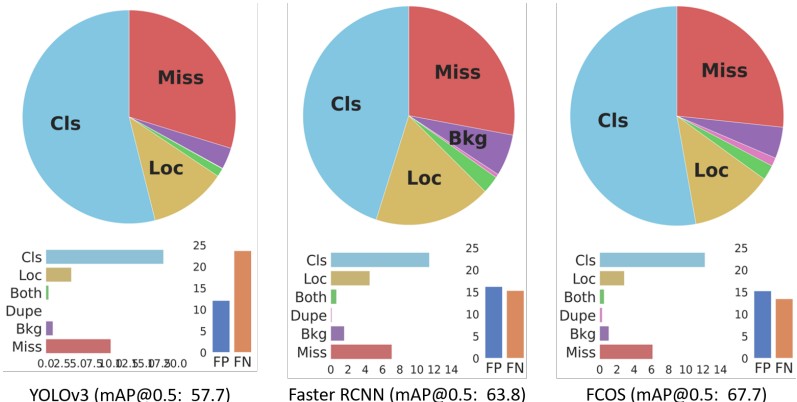

Figure 12: **Breakdown of Detection Errors on ImageNet-VID using TIDE [37].** We can observe that classification errors dominate in this dataset, when compared to error breakdowns on COCO.

Table 9: **Overhead of Context.** While the context is obtained only when the controller is called to make a new runtime decision, there is an associated computational overhead. The overhead includes the runtime for the controller.

| Scale | No Context | AdaScale | AdaScale + Scene Agg. | AdaScale + Scene Agg. + Switch. |
|-------|-----------|----------|----------------------|-------------------------------|
| 900 | 104.1 ms | 19ms | 19.6ms | 39.2ms |
| 600 | 55.7ms | 9.5ms | 9.7ms | 18.5ms |
| 300 | 1.3ms | 32.2ms | 3.5ms | 7.3ms |

assign the switchability category to either, *low, medium, or high*. The *low* switchability category indicates that all the models (both expensive and inexpensive models) have similar performance and hence model switching is not particularly useful. Whereas, a *high* switchability category indicates switching will result in better accuracy. Finally, we train the classifier in a supervised manner, with input features from the backbone.

**Adaptive Scale Regressor.** We followed the method detailed in [2]. The regression target for an image is derived by comparing foreground box predictions at different scales (E.g., in our case s={720,600,480,360,240}). Common foreground objects have the minimum loss at the selected scale. We train a regressor in a supervised manner using these scale labels.

**Contention Sensor.** The contention generator is a CUDA kernel that performs arithmetic on arrays, following [4]. The contention levels are discrete in nature measured using `nvidia-smi` and `tegrastats` device as appropriate. Change in contention levels influences the latency of the detector as GPU cores are used by the generator.

**Overhead of Context.** Table 9 shows that the overhead of context for a single frame (presented numbers are averaged over 1000 distinct frames) at different scales is around 25-35%. The majority of the overhead is from the AdaScale regressor and switchability classifier. However, it should be noted that metrics are computed only when the controller is executed (i.e., once every 30 detection calls), hence, amortized overhead per frame is merely around 1% during runtime.

## D  Additional Details on Learning the Controller

To learn $Q_\theta$, loss $\mathcal{L}$ takes into account all the decision dimensions. $\mathcal{L}$ is calculated by sampling batch of {context, action and reward} tuples, i.e., $(z, a, r)$ from the controller's replay buffer $B$,

$$\mathcal{L} = \mathbb{E}_{(z,a,r)\sim B}\left[\frac{1}{|\mathbb{D}|}\sum_{a_i\in a}[r - Q_\theta(z, a_i)]^2\right]$$

---

**Algorithm 3:** Modified Shrinking Tail Scheduler

---

1: Given finishing time $s$ and algorithm runtimes for different configurations $\rho$ in the unit of frames (assuming $\rho[i] > 1 \ \forall \ i$) and current algorithm configuration $a$, this policy returns whether the algorithm should wait for the next frame.
2: Define tail function $\tau(t) = t - \lfloor t \rfloor$
3: **return** $[\tau(s + \rho[a]) < \tau(s)]$ (Iverson bracket)

---

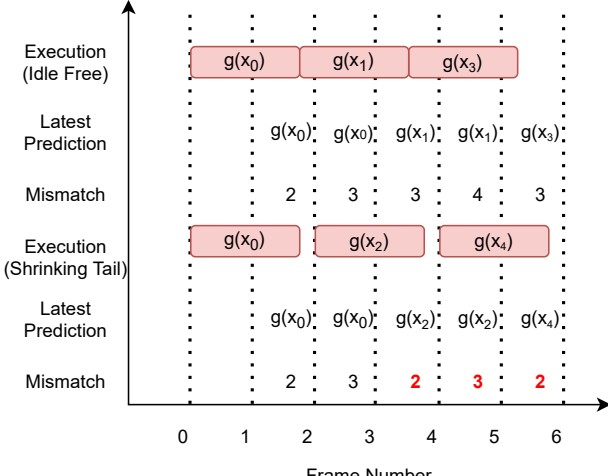

Figure 13: Temporal mismatch between the latest available prediction and the current frame in the stream employing Idle Free and Shrinking Tail scheduling of a function $g$. Lower mismatch (depicted in red) is better.

To perform exploration-exploitation during training, we employed UCB and $\epsilon$-greedy strategies. We empirically observed $\epsilon$-greedy converges faster and is our default strategy. While testing our learned policies, we act upon actions which provide the maximum expected reward given input context.

**UCB:** Action is chosen using the following equation,

$$a_\tau = \left\{ \underset{d \in D_i}{\mathrm{argmax}} \left[ Q_\theta(z_\tau, d) + c\sqrt{\frac{ln(\tau)}{N_\tau(d)}} \right] \ \Big| \ \forall D_i \in \mathbb{D} \right\} \tag{3}$$

where $\tau$ denotes the number of trials. $N_\tau(d)$ denotes the number of times that action $d$ has been selected prior to step $\tau$, and the parameter $c > 0$ determines the confidence level.

$\epsilon$**-Greedy:** We set initial $\epsilon$ as 1, and decay it by a factor till it reaches a minimum threshold. Action is chosen using via,

$$a_\tau = \begin{cases} \{d_i \sim U(D_i) \ \ | \ \ \forall D_i \in D\} & \text{with probability } \epsilon \\ \left\{ \underset{d \in D_i}{\mathrm{argmax}} Q_\theta(z_\tau, d) \ \ | \ \ \forall D_i \in D \right\} & \text{otherwise} \end{cases} \tag{4}$$

where $U$ is uniform distribution.

## E   Modified Shrinking Tail Scheduler

We discuss the difference between the scheduling algorithms that were mentioned in the main text. Figure 13 depicts temporal mismatch, the current frame number that has been recieved compared to the frame currently being processed. As expected, we would like mismatch values to be as close to zero as possible. Let's assume that the runtime of a detector is 60ms. We receive a new frame every 33ms. Say, we received frame 0 at t = 0ms, and started processing. Thus, we wait till 60ms for the result of Frame 0 by the time we have received Frame 1 for 27ms. At this point, we can make one of two choices, call the detector on Frame 1 (**idle-free scheduling**), or wait for 6ms to receive Frame 2

and then make the blocking call (**shrinking tail scheduling**). Turns out, it's better to wait as that reduces the temporal mismatch [6].

As noted in Section 3.5, shrinking tail scheduler [6] assumes algorithm runtime is constant, which is not true in our case. However, the space of configurations is *discrete*, and runtime for *every* configuration can be assumed to be constant. Algorithm 3 incorporates that assumption.

# F  Deployment Efficiency.

Deployment efficiency is a critical factor when deploying to edge devices. Let us consider the scenario where our Chanakya has to learn tradeoffs for two dimensions, $S = |D_s|$, $P = |D_{np}|$. We denote number of training epochs as $n_{epochs}$, number of training images as $N_{train}$, number of validation images as $N_{val}$, a scaling factor to accommodate for streaming scenarios $\beta$ where $\beta \geq 1$, matrices $M_{prob} \in \mathbb{R}^{S \times P}$ indicating the probability of an action chosen and $M_{lat} \in \mathbb{R}^{S \times P}$ indicating latency. Time required for training Chanakya and other baselines can be formulated as: $Chanakya_{time} = \frac{M_{prob} \cdot M_{lat}}{\beta}(n_{epochs} \cdot N_{train})$. For static policy such as Streamer is $Static_{time} = N_{train} \cdot \sum_{i \in D_s, j \in D_{np}} M_{lat}$ and for dynamic-offline policy such as AdaScale, it is $Dynamic_{time} = \frac{N_{train} \cdot \sum_{i \in D_{scale}} M_{lat}}{\beta}$.

We can now compute deployment efficiency of our approach against static and dynamic policies as:

$$\eta_1 = \frac{Static_{time}}{Chanakya_{time}} = \frac{\sum_{i \in D_s, j \in D_{np}} M_{lat}}{M_{prob} \cdot M_{lat} \cdot n_{epochs}}$$

$$\eta_2 = \frac{Dynamic_{time}}{Chanakya_{time}} = \frac{\sum_{i \in D_s} M_{lat}}{M_{prob} \cdot M_{lat} \cdot n_{epochs} \cdot \beta}$$

On Argoverse-HD containing approximately 55K images, Chanakya is 3.46x and 2.74x faster on the P40 server GPU and the Jetson Xavier NX respectively. The efficiency is slightly lower on the Jetson due to increased number of epochs during training. In case of dynamic-offline policy (AdaScale), since only one forward pass is required for all scales, AdaScale is twice as efficient as ours, but performs extremely poorly. Furthermore, deployment efficiency only increases as the configuration space combinatorially increases.

# G  Application Scenarios

We briefly discuss two application scenarios, and observe how context and decisions change.

Consider a smartphone-based driver assistant [46, 47] employing vehicle detection and tracking. Process contention (from other mobile applications) and image content metrics form the context. We can assume the smartphone is deriving energy from the car and the expected response time is too low to perform remote execution on the cloud. Thus the decision dimensions are – detector resolution, tracker resolution and temporal stride, meta-parameters like number of proposals.

Consider another scenario, say, a drone survey platform [48, 49, 50] tasked with search and rescue operations. The drones have cloud support at the ground station along with onboard compute. The platform executes real-time person detection and tracking in the surveyed area. Here, the context is image content, energy use, device temperature, and dynamic network conditions, but not process contention. The decision dimension now includes remote execution on the cloud. The decision is thus affected by the context in new ways, for instance, if the decision is to execute to cloud machine, the detector choice and scale would now depend on network conditions [35].

For driver assistance application, developers need to decide the detector and tracker model, their scale, detector's number of proposals, and tracker stride. Context provided to the decision function include the contention levels from other running processes apart from context derived from image content. Drone survey platform would have an additional binary decision to offload frames to the GPU on the edge-cloud ground station. Such additions can be accommodated in our decisions without any changes in training or scheduling algorithms. We can provide additional context (network conditions: latency and bandwidth, power etc) to Chanakya. Chanakya can be trained without changes for novel real-time perception scenarios (unlike [1, 34, 4, 32, 5]), highlighting its flexibility and adaptablity.

Adopting Chanakya simplifies development cycle for real-time perception by letting developers focus on identifying and implementing context, and defining decisions which introduce computational tradeoffs. The developers provide models, the decision space, and a dataset for training Chanakya. Chanakya would automatically learn the decision function depending on the context and execute decisions with favorable tradeoffs. Chanakya imposes no restriction on task and can be applied to any task (detection, segmentation etc).