# OpenReview forum: "Chanakya: Learning Runtime Decisions for Adaptive Real-Time Perception"
_NeurIPS.cc/2023/Conference — NeurIPS 2023 poster_

### Official Review · Reviewer_6nGS · 2023-07-06

**Soundness:** 3 good
**Presentation:** 3 good
**Contribution:** 3 good
**Rating:** 6
**Confidence:** 3

**Summary:**

This paper proposes a learned approximate execution framework named Chanakya. Chanakya considers intrinsic context like images and extrinsic context like latency and predicts runtime decisions. The reward function helps to learn a better trade-off online. Extensive experiments on the Argoverse-HD dataset show that Chanakya outperforms static policy and can be applied to new hardware and action spaces.

**Strengths:**

1. This paper jointly optimizes accuracy and latency for real-time tasks. The reward function proposed considers both the simultaneous optimization between accuracy and latency, and different characteristics of the video sequence.
2. The detailed analysis of experiment results demonstrates the performance improvements of Chanakya.
3. This paper is well-organized and well-written to read.

**Weaknesses:**

1. Some settings of experiments are not clarified, like offline upper bound and RL training method.
2. Some details of learning the controller should be introduced in the main part of the paper.

**Questions:**

1. Why did employing L_{streaming} not converge which is mentioned in Line 165?
2. In table 1, how do you get sAP for offline upper bound?

**Limitations:**

The authors just mention that their method might be employed to improve unethical applications.

---

> ### Author Rebuttal · Authors · 2023-08-06
>
> Thank you for your thought provoking review, we shall incorporate the suggested point in the final manuscript!
> 1. Thank you for the upper bound suggestion, please see main comment.
> 2. Details of controller training method are provided in the supplementary along with the anonymous code link, please let us know if there are specific questions that need clarification.
> 3. In both the datasets, the number of videos available for training is of the magnitude of hundreds (say N). L_{streaming} yields only one reward value per video execution (thus one epoch → N samples for training) and the model performance was worse than static policies after 5 epochs. To circumvent this, we essentially chunked every video into temporal segments (essentially R_{1}) yielding far more samples for training ((T/30)*N samples where T is the average length of the video).

---

> > ### Comment · Reviewer_6nGS · 2023-08-17
> >
> > Thanks for the response. I keep my score as 6.

---

### Official Review · Reviewer_7j3D · 2023-07-07

**Soundness:** 2 fair
**Presentation:** 3 good
**Contribution:** 2 fair
**Rating:** 6
**Confidence:** 3

**Summary:**

This paper introduces Chanakya, a framework for computation planning in real-time (streaming) perception. Chanakya uses a novel reward to make run-time decisions based on content- and system-based characteristics, simultaneously optimizing for both accuracy and latency. The proposed controller is learned on individual frames and uses a normalized reward to reduce bias. Chanakya’s performance is evaluated on a real-time perception stack consisting of a detector, scheduler, forecaster and tracker; these individual components are made part of the search space (eg: choice of model), along with their associated configurations (scale, #proposals, etc.). Results are reported across edge devices, runtime contention levels, and for cases where the perception stack has been migrated from one GPU to another.


**Strengths:**

* This is a well-written paper, and the core ideas have been explained clearly.
* Impact & relevance: the paper targets real-time perception, which is an important component of modern AV and embodied systems.
* Strong results: Chanakya appears to outperform SOTA static and dynamic approaches, and ones designed by domain experts.


**Weaknesses:**

* It is not clear how well Chanakya scales with increasing search space size. While the paper includes an evaluation of a reasonable number of decision dimensions, it would be interesting to get an understanding of Chanakya's scalability limits, especially since the authors claim in Section 2 that the search space could potentially be expanded by trying to account for varying input resolutions or via techniques such as pruning and/or quantization.
* No comparisons to non-RL-based approaches, including rule-based and purely latency-focused systems.


**Questions:**

The comparison in Figure 1(c) is to RetinaNet, which is a network from 2017. Do more modern networks (eg: ConvNeXt/Swin, CoAt-Net, MaxViT, etc.) fulfill the real-time/streaming constraint by being able to handle higher resolutions?

**Limitations:**

Limitations have been adequately addressed in the paper.

---

> ### Author Rebuttal · Authors · 2023-08-06
>
> Thank you for your insightful review, we shall incorporate these points in the final version of the manuscript.
> 1. **Increasing Search Space:** We agree that this is an important direction. However, adding even more decision dimensions – such as edge-cloud processing (should an image be sent to a cloud GPU or executed at edge GPU like Jetson) would be involve development of more complicated simulations (for network variability). We leave this for future work.
> 2. **Prior work:** The execution model of prior work is different. It’s difficult to compare as every algorithm needs to be modified for a fair comparison. We did include one such baseline AdaScale [2] (and modified the scheduling algorithm) – which is not RL based, which reduces image scale and solely optimizes latency by proxy. Performance is worse by 35%.
> 3. **New Models:** We are looking to develop execution policies that are agnostic to the model choice and hardware choice. Recent work has shown models with better accuracy-latency tradeoffs (like, StreamYOLO, YOLO-X), however, the fundamental tradeoff remains. Models like StreamYOLO do satisfy the real-time constraint on high end GPUs like V100, but are not hardware-agnostic, due to design limitations performance suffers on older and edge hardware [13].

---

> > ### Comment · Reviewer_7j3D · 2023-08-21
> >
> > Thank you for the response. I will keep my score the same.

---

### Official Review · Reviewer_Gsqp · 2023-07-07

**Soundness:** 2 fair
**Presentation:** 3 good
**Contribution:** 2 fair
**Rating:** 6
**Confidence:** 4

**Summary:**

The paper looks at the problem of unpredictable compute requirements for real-time perception. It addresses this in term of a multi-objective optimisation problem (quality of results and latency). The authors propose using RL in order to optimise the selection of various characteristics in order to achieve an optimal result.

**Strengths:**

The paper is well written and provides a good introduction for the reader and guides them through the work.

**Weaknesses:**

The proposed approach does not come across as that novel. There is a lot of prior use of RL techniques for multi-objective optimisation which the authors don't appear to have looked at.

**Questions:**

- The authors spend a lot of the introduction naming and saying how good their approach is. This should be more analytical than what is presented.

- Lines 89-94 give a formal definition of the perception problem. However, most of the introduced terminology does not get used further in the paper. This space could be used for other material.

- Figures 2,3&4 present results, but it is not clear where these results came from and how they were produced.

- "In our experiments we observed that controller collapsed to predict a single configuration when rewards of this form were provided (depending on λ), and did not learn" - which experiments?

-

**Limitations:**

Small amount of discussion of limitations in conclusion.

---

> ### Author Rebuttal · Authors · 2023-08-06
>
> Thank you for your suggestions in the review, they will definitely improve the quality of the manuscript.
> 1. Novelty: Please see main comment.
> 2. Thank you for this suggestion. We have added some key numerical highlights in main comment, which we will add in the introduction.
> 3. Thank you for this suggestion, we shall add a connection to Section 3.4. The setup mentioned in Section 3.1 is necessary for understanding Section 3.4.
> 4. We have mentioned experimental, implementation and a few algorithmic details in supplementary, we shall add more explanation there. 5. Please let us know what exactly is missing in the results section, we shall do our best to incorporate the suggestion.
> 6. Apologies for the confusion, we trained the controller with reward defined in [3] and the learnt policy always picked a singleton configuration (like static policy) depending on the hyperparameter \lambda. Previous work [3, 21] data is not public, so we could not compare directly on their data. Hence, we omitted the comparison in our results.

---

> > ### Comment · Reviewer_Gsqp · 2023-08-20
> > **Thanks for the comments**
> >
> > The authors have addressed a number of the concerns that I had. However, they have not fully addressed the underlying issues. As such I feel my mark should stand.

---

### Official Review · Reviewer_fYnV · 2023-07-14

**Soundness:** 3 good
**Presentation:** 3 good
**Contribution:** 3 good
**Rating:** 5
**Confidence:** 2

**Summary:**

 This work provides a novel learning-based approximate execution framework to learn runtime decisions for real-time perception. The learned controller proves to be efficient and performant, which appears to be useful for many real-time perception applications in the cloud and edge.

**Strengths:**

1. The focus of the work on real-time perception system's runtime execution decisions is important and timely.
2. The proposed Chanakya proves to learn performant execution policies and can incorporate new decision dimensions and be ported to different hardwares.

**Weaknesses:**

The experimental results are only reported on detection tasks, and other perception task like segmentation might be reported.
The decision dimention seems to be discrete and limited.

**Questions:**

How to evaluate and compare between the controllers on different hardware, which can have different features?

**Limitations:**

As listed in weakness part, more experimental results on other perception task can be reported, and not sure whether the decision space is valid.

---

> ### Author Rebuttal · Authors · 2023-08-06
>
> Thank you for your insightful review, we are glad to see that you agree that Chanakya has been proved to learn performant execution policies in a variety of scenarios.
>
> 1. **Single Task Experiment:** We performed a thorough study using one task – detection, but across datasets, scenarios and edge and cloud devices. Critically, runtime characteristics of instance segmentation models (e.g. Mask R-CNN) are like detection models (e.g. Faster R-CNN) [6]. For experimenting with instance segmentation, we need to additionally define different set of context functions and a more complicated forecasting mechanism which further complicates experimental design.
> 2. **Discrete Dimensions:** Most of the prior work also operate on discrete decision dimension(s), usually one such dimension.
> 3. **Evaluation across hardware:** Could you please clarify the question regarding comparing controllers on different hardware? What is to be compared?

---

> > ### Comment · Reviewer_fYnV · 2023-08-17
> >
> > Thanks and I will keep the score as is.

---

### Official Review · Reviewer_L1Mt · 2023-07-24

**Soundness:** 3 good
**Presentation:** 2 fair
**Contribution:** 2 fair
**Rating:** 5
**Confidence:** 1

**Summary:**

The author designs Chanakya, a learned execution framework for streaming perception that jointly optimizes accuracy and latency. To achieve so, the framework captures both intrinsic and extrinsic contexts and utilizes a novel reward function to train a learned controller.

**Strengths:**

1. The motivation is clear and the discussion of related work is comprehensive.
2. The description of experimental setting is detailed and results of ablation studies are provided.

**Weaknesses:**

1. Novelty is somewhat limited. The methodological innovation in this paper is limited to use both intrinsic and extrinsic contexts to better learn runtime decisions with reinforcement learning based methods. The fundamental methodology is mostly shared with prior work.
2. Presentation can be improved. Table 1 is at the top of page 6, but no context related to it appears until page 7. Asynchronous Training and Training for edge devices in Section 3.5 seems unnecessary to discuss.
3. Baselines are incomplete. I think the experiment should at least have vanilla RL-based baselines and then show the effectiveness of the proposed framework step by step.
4. I'm not sure it's a good idea to incorporate all factors into a learning controller. For single GPU program, I believe rule-based decision algorithms can adapt extrinsic factors (e.g. software and hardware status) very well.

**Questions:**

How does your approach compare to hybrid solutions that employ handcrafted rule-based heuristics for extrinsic factors and learning-based decision algorithms for intrinsic factors?

**Limitations:**

In the subsection "Limitations and Broader Impact", the authors point out that their framework can be used to deploy unethical applications, and I think this is just a potential negative societal impact, not a limitation. Therefore, I encourage authors to talk about the limitations of your work and potential solutions.

---

> ### Author Rebuttal · Authors · 2023-08-06
>
> Thank you for your insightful review, it has helped us clarify aspects of our work. We believe the raised concerns are addressable in the manuscript.
> 1. Please see the main comment for novelty concerns.
> 2. We shall improve the presentation of the work and move the tables and add a high level overview diagram.
> 3. **Vanilla RL:** Please see main comment.
> 4. **All Factors:** This is a problem specifically on edge devices, where users have multiple applications running on device simultaneously. While rule based decisions for this consideration do appear to work [4], however, even methods like ApproxDet hypothesize using NNs to infer configurations and RL to learn scheduling due to its distinct advantages.
> 5. **Hybrid Methods:**
>      - We compare with one such baseline for heuristics with a learned metric – AdaScale [2]. The rule is greedily accepting the scale value from the learned scale predictor for the next frame.
>      - No other entirely learning-based real-time decision algorithms or “handcrafted rule-based heuristics for extrinsic factors and learning-based decision algorithms for intrinsic” exist as far as we are aware. We present first such general framework.
>
> 6. **Limitations:** (a) Our training and test environment is the same and we haven’t tested the robustness (and adverserial robustness) of the learnt policy. With the vast literature on RL in domain/environment randomization, we expect that this limitation will be resolved in future work. (b) We considered context to be instantaneous w.r.t to a frame, however, for some considerations like power and temperature, accounting for context over a longer time horizon is generally required [Chen2021].
>
> [Chen2021] Enforcing Policy Feasibility Constraints through Differentiable Projection for Energy Optimization, e-Energy, 2021

---

> > ### Comment · Reviewer_L1Mt · 2023-08-19
> >
> > Thanks to the authors and other reviewers for their replies and discussions. I read them carefully and it helped me understand the work better. I raised the rating to 5 and lowered the confidence to 1.

---

### Official Review · Reviewer_Y14P · 2023-07-30

**Soundness:** 3 good
**Presentation:** 2 fair
**Contribution:** 2 fair
**Rating:** 5
**Confidence:** 3

**Summary:**

The paper presents Chanakya, a learned framework for real-time perception that automatically balances accuracy and latency. Unlike previous fixed rule-based methods, Chanakya considers both intrinsic and extrinsic factors and is trained to make flexible decisions. evaluations show that it outperforms existing execution policies on both server GPUs and edge devices with low overhead.

**Strengths:**

1. It proposes a learning based approximate execution framework to learn runtime decisions such as the resolutions of input, which model to use, etc.
2. The proposed framework considers multiple context both intrinsic and extrinsic to improve the reliability of the runtime decisions.

**Weaknesses:**

1. It is unclear how generalizable is the trained policy. More discussions on when train from scratch and when transfer learning and the corresponding cost when the system is in a different environment is needed.

**Questions:**

1. It would be good to have a diagram figure to show the high level overview of the whole framework.
2. There exists some work that use elastic model setting for different scenarios such as [1]. More discussions and comparisons are needed.

[1] Wang, Chien-Yao, Alexey Bochkovskiy, and Hong-Yuan Mark Liao. "Scaled-yolov4: Scaling cross stage partial network." Proceedings of the IEEE/cvf conference on computer vision and pattern recognition. 2021.

---

> ### Author Rebuttal · Authors · 2023-08-06
>
> Thank you for your insightful review!
>
> 1. We train the controller from scratch depending on the conditions simulated. We leave robustness as future work, as large number of inferences from existing RL works can be drawn to improve this aspect (such as domain randomization). Cost of training the policy is a fraction of training the detection models themselves (a 4 layer MLP for 10 epochs).
>
> 2. Thank you for this suggestion! We shall add a high-level overview diagram in the camera ready version of the paper.
>
> 3. The paper (Scaled-YOLOv4) suggested is interesting, structure of the model is search for apart from network depth and width to obtain a family of models with good accuracy-latency characteristics. Critically, that paper optimizes decisions taken during training a family of detectors, which is not our focus. We focus only on runtime decisions and we can learn to select from a slew of already trained models (such as family in the mentioned paper) for a given hardware/environment. We shall mention it in related work.

---

> > ### Comment · Reviewer_Y14P · 2023-08-20
> >
> > I appreciate the answers from the authors. I'd like to keep my score.

---

### Author Rebuttal · Authors · 2023-08-06

We thank all the reviewers for their insightful comments. We are glad to see that reviewers noted that our paper is “well written” (R4, R5, R6) and “well-organized” (R6) as we tackled an “relevant and impactful problem” (R5) with a “clear motivation” (R3). It is heartening to note that our “learned execution framework for real-time perception” is “proved to learned performant execution policies” (R4) through “detailed experimental setting” (R2), showing “strong results” (R5) by “demonstrating the performance improvements” (R6) while considering “both intrinsic and extrinsic” (R1) environment contexts.

### Contribution
We propose a novel learned framework to learn execution decisions for real-time perception, optimizing accuracy and latency simultaneously in a variety of scenarios. Detector/tracker etc are **not retrained**, improvements are **only from learned execution policy**.
1. We obtain improvements of 17% on Argoverse-HD and 9% on Imagenet-VID for a pre-defined perception system. This improvement is complementary to the components and is better than the best static policies found by benchmarking and domain experts.
2. Our method can improve performance in a variety of hardware and scenarios. E.g., when environment has process contention, our execution policy degrades better than static policies. Our algorithms (like scheduler) are not modified to account for new considerations, unlike prior work.

### Novelty (R2/R4)
While finding good runtime decisions through execution frameworks, and improving the possible pareto frontier through better models for real-time perception has been studied, however,
1. Prior work do not learn the execution framework and thus generally operate on **one** consideration or decision dimension – either intrinsic or extrinsic. In reality, real-time perception systems have many considerations. Thus, they need to modify algorithms, e.g. the scheduler to incorporate any new consideration (and model the interactions of every consideration with others; or come up with a new heuristic). We claim novelty and show advantages of our learned execution framework.
2. Prior work optimize suboptimal proxy metrics. For example, optimizing scale solely [2] (a proxy for latency itself) drastically reduced performance. We claim novelty of the objective to optimize and the design decision to situate our execution framework in the streaming perception problem [6] which led to this natural objective.

### RL Novelty/Vanilla RL (R2/R4)
Proposing new RL algorithms is not the focus of this work and there might be other RL algorithms (like Actor-Critic Methods) that are also applicable. Due to resource constrained nature of real-time perception, we formulate an RL setup that does not hamper performance over static policies.

### Offline Upper Bound (R6)
We take the best configuration of the static policy from [6] and simulate a detector with 0ms latency for every frame and obtain an upper bound. We shall mention this in the camera ready version.

Denoting R1 to 6 for Y14P, L1Mt, fYnV, Gsqp, 7j3D, 6nGS respectively.

---

### Decision · Program_Chairs · 2023-09-21

**Decision:**

Accept (poster)

**Comment:**

The submission proposes a method for optimizing resource utilization when running neural networks for perception tasks. In particular, two considered tradeoff factors are accuracy and latency, which are important for real-world applications, such as AVs. The method uses an approximate execution framework to automatically learn decisions induced by these tradeoffs, which incorporates the so-called intrinsic (eg scene clutter) and extrinsic (eg hardware status) characteristics. The reviewers unanimously recommend the submission for acceptance. The weaknesses raised in the reviews were relatively on the insignificant side, and the clarifications provided in the rebuttal were found helpful. The authors are recommended to incorporate them and the other comments raised in the reviews in preparing their camera ready.